©c Author(s) 2020. CC BY 4.0 License.





# Biomass burning aerosols in the southern hemispheric midlatitudes as observed with a multiwavelength polarization Raman lidar

Athena Augusta Floutsi[1], Holger Baars[1], Martin Radenz[1], Moritz Haarig[1], Zhenping Yin[1,2,3],
Patric Seifert[1], Cristofer Jimenez[1], Ulla Wandinger[1], Ronny Engelmann[1], Boris Barja[4],
Felix Zamorano[4], and Albert Ansmann[1]

[1]Leibniz Institute for Tropospheric Research, Leipzig, Germany
[2]School of Electronic Information, Wuhan University, Wuhan, China
[3]Key Laboratory of Geospace Environment and Geodesy, Ministry of Education, Wuhan, China
[4]Atmospheric Research Laboratory, University of Magallanes, Punta Arenas, Chile

**Correspondence:** Athena Augusta Floutsi (floutsi@tropos.de)

**Abstract.** In this paper, we present long-term observations of the multiwavelength Raman lidar Polly$^{XT}$ conducted in the framework of the DACAPO-PESO campaign. Regardless the relatively clean atmosphere in the southern mid-latitude oceans region, we observed regularly events of long-range transported smoke, originating either from regional sources in South America or from Australia. Two case studies will be discussed. Both cases were identified as smoke events and they occurred on 5 February 2019 and 11 March 2019. For the first case considered, the lofted smoke layer was located at an altitude between 1 and 4.2 km, and apart from the predominance of smoke particles, particle linear depolarization values indicated the presence of dust particles in the layer. Mean lidar ratio values at 355 and 532 nm were $49 \pm 12$ and $24 \pm 18$ sr respectively, while the mean particle linear depolarization was $7.6 \pm 3.6$ % at 532 nm. The advection of smoke and dust particles above Punta Arenas affected significantly the available CCN and INP in the lower troposphere, and triggered effectively ice crystal formation processes. Regarding the second case, the thin smoke layers were observed at altitudes between 5.5–7, 9 and 11 km. The particle linear depolarization ratio at 532 nm increased rapidly with height, starting from 2% for the lowest two layers and increasing up to 9.5% for the highest layer, indicating the possible presence of non-spherical coated soot aggregates. INP activation was effectively facilitated. The long-term analysis of the one year of observations showed that tropospheric smoke advection over Punta Arenas occurred 16 times (lasting from 1 to 17 hours), regularly distributed over the period and with high potential to influence cloud formation in the otherwise pristine environment of the region.

## 1 Introduction

Punta Arenas (53.2°S, 70.9°W) is located at the Strait of Magellan, at the southern tip of South America (location indicated with yellow star in Fig. 1). Constant westerly to north-westerly winds prevail through the whole year induced by the Antarctic low-pressure belt (Schneider et al., 2003). Since it is located in the most southerly continental area (with the exception of Antarctica), with New Zealand, Tasmania and Australia being the nearest (>8.000 km) landmasses towards the direction of the prevailing winds, clean pristine marine air masses dominate the aerosol conditions. The aforementioned geographical and





climatic conditions make Punta Arenas an ideal test-bed for studies of aerosol, aerosol-cloud and radiation interactions under rather clean conditions with low anthropogenic impact (Kanitz et al., 2011, 2014; Foth et al., 2019).

In the southern mid-latitudes, there is a lack of long-term ground-based remote-sensing observations of aerosols and clouds,
mainly due to the absence of substantial land masses below 40 degrees latitude. Immler and Schrems (2002) used the pristine location of Punta Arenas to contrast the properties of mid-latitude cirrus clouds from those observed in the aerosol-burden northern midlatitudes. First lidar-based studies of the vertical aerosol distribution over Punta Arenas have been performed in the frame of the Aerosol Lidar measurement at Punta Arenas in the frame of Chilean germAn cooperation (ALPACA) campaign (Kanitz et al., 2013; Baars et al., 2016; Foth et al., 2019). The portable Raman lidar Polly$^{XT}$ (Althausen et al., 2009;
Engelmann et al., 2016) was deployed at the Magallanes University in Punta Arenas and operated continuously for 4 months (4 Dec. 2009 – 4 Apr. 2010). During the ALPACA campaign, pristine marine conditions dominated the measurements, while lofted aerosol layers were observed only eight times. From these eight lofted layers examined, seven of them had an aerosol optical thickness less than 0.05. The origins of these layers were mainly bush fires in Australia and Patagonian dust (Foth et al., 2019).
As a follow-up of ALPACA, the Dynamics, Aerosol, Cloud and Precipitation Observations in the Pristine Environment of the Southern Ocean (DACAPO-PESO) campaign started in November 2018 as a collaboration between the Leibniz Institute for Troposperic Research (TROPOS), the Leipzig Institute for Meteorology (LIM) and the University of Magallanes (UMAG) in Punta Arenas, Chile. DACAPO-PESO is focused on the investigation of cloud formation and aerosol-cloud interaction in environments of contrasting aerosol conditions and relies mainly on synergistic algorithms using state-of-the-art lidar and
radar techniques (more information in Sec. 2). Within this publication, we present lidar observations of the vertical aerosol distribution above Punta Arenas during two interesting events that took place during the DACAPO-PESO campaign. These events demonstrate distinctively different lofted aerosol layers (in terms of physical and optical properties) in the clean and pristine environment of Punta Arenas. Since aerosol particles influence the evolution, lifetime, and microphysical properties of clouds by acting as cloud condensation nuclei (CCN) and ice nucleating particles (INP), CCN and INP concentrations were
derived from the lidar signals for the corresponding cases.

This paper is organized as follows: Section 2 provides information on the experiment, our data sources, instrumentation and methodology. Section 3 presents our analysis for two different smoke events as observed over the site of Punta Arenas, focusing on the optical properties of the aerosol particles. Section 4 gives an overview of all the lofted layers that were observed, above our measurement site, during 2019. Section 5 discusses the main findings and highlights the most important conclusions of this
study.

## 2  Experiment and Instrumentation

The DACAPO-PESO (https://dacapo.tropos.de/) campaign started in November 2018 and will continue until June 2020. This 19 month campaign is focused on aerosol, clouds, ice nucleation particles and aerosol-cloud dynamics relationships in the pristine environment of the southern-hemisphere midlatitudes. These complex topics need a synergistic approach in terms of



instrumentation and therefore the Leipzig Aerosol and Cloud Remote Observations System (LACROS, Bühl et al. (2013)) was deployed on site (53.2°S, 70.9°W, 9 m a.s.l.). LACROS comprises a unique set of active and passive remote-sensing instruments which are to a large extent containerized (Fig. 1). More specifically, some key components are a 35 GHz cloud radar, a Raman and depolarization lidar as part of PollyNET (Baars et al., 2016), a 2-µm scanning Doppler lidar, a 24-GHz micro rain radar, a microwave radiometer, a Sun photometer, a disdrometer and a radiation station with an all-sky camera. In

addition, a 94-GHz cloud radar (provided by LIM) was also deployed until 1 October 2019. A Sun photometer was installed on the roof platform of the main building of UMAG, as part of the Aerosol Robotic Network (AERONET, Holben et al. (1998)). The LACROS observations are accompanied by regular observations of a multi-wavelength lidar of UMAG, which is operated at the site since 2016 in the frame of the South American Environmental Risk Management Network (SAVER-Net) and the Latin-American Lidar Network (LALINET) (Ristori et al., 2018).

## 2.1   Lidar measurements

In this paper, we will focus on atmospheric measurements which were conducted with the Raman lidar Polly$^{XT}$ (Engelmann et al., 2016; Baars et al., 2016). Polly$^{XT}$ utilizes a Nd:YAG laser that emits light at 3 different wavelengths, that are 355, 532 and 1064 nm. The receiver consists of 12 channels and enables measurements of elastically (355, 532, 1064 nm) and Raman scattered light (387, 607 nm for aerosols and 407 nm for water vapour), depolarization state of the backscatter light (at 355 and

532 nm). A near-range telescope allows the detection of 355 nm, 387 nm, 532 nm, and 607 nm scatter light from about 60-80 m above ground. Vertical profiles can be obtained with an uppermost detection height of around 20 km a.g.l. (above ground level). Data from all channels are acquired with a vertical resolution of 7.5 m and a temporal resolution of 30 s. The extinction and backscatter coefficients were determined as described by Ansmann and Müller (2005), while the depolarization ratio was determined according to standard procedures applied in the European lidar network EARLINET (D'Amico et al., 2015; Mattis

et al., 2016). Quality assurance (e.g.Wandinger et al. (2016); Freudenthaler (2016); Belegante et al. (2018)) is a key aspect, even when Polly lidar systems are operated outside Europe. Regarding the signal analysis, the influence of noise to the lidar signals was reduced by means of a vertical gliding averaging filter. Window lengths varied for the two cases considered, such as not to lose any important structural information through the process of averaging. For the first case a window length of 562.5 m was applied for the backscatter coefficient, Ångström exponent and particle linear depolarization ratio, while a window

length of 1492.5 m was applied for the much noisier lidar ratio. For the second case, a window length of 307.5 m was used for the backscatter coefficient and the particle linear depolarization ratio profiles, and a length of 1492.5 m was applied to the Ångström exponent.

   The presented CCN and INP concentrations have been derived following the methodology proposed in Mamouri and Ansmann (2016). Firstly the dust and smoke contribution was separated using the particle linear depolarization ratio. A lidar ratio

of 50 sr for smoke and 55 sr for dust was used to obtain the extinction coefficient per aerosol type. The smoke-related extinction coefficient is transferred to the number concentration $n_{50,c}$ of particles with dry radius larger than 50 nm (for CCN) and



the surface area concentration $s_c$ (for INP) using the following equations:

$$n_{50,c} = c_{60,c} \times \sigma_c^{\chi_c} \tag{1}$$

$$s_c = c_{s,c} \times \sigma_c \tag{2}$$

with $c_{60,c} = 25.3$ cm$^{-3}$ for $\sigma =1$ Mm$^{-1}$ , $\chi_c = 0.94$ and $c_{s,c} = 2.80 \times 10^{-12}$ Mm $\times$ m$^2$ $\times$ cm$^{-3}$ at 532 nm based on long-term Aeronet observations over Leipzig (Mamouri and Ansmann, 2016). To reduce the uncertainties, a specific parameterization for smoke is under development. All soot particles with a dry diameter larger than 50 nm are assumed to activate as CCN at a supersaturation of 0.2%. More specifically, the INP parameterizations and input parameters for soot and dust are from Ullrich

et al. (2017), in accordance with the temperature regime of each case studied. Measurement quicklooks for the whole campaign period are available at http://polly.tropos.de in near real-time.

## 2.2  Auxiliary data

The Sun photometer located at UMAG (53.1°S, 70.9°W, 25 m a.s.l., site name: Punta_Arenas_UMAG), measures the column-integrated extinction coefficient at 8 channels from 340 to 1640 nm. Level 1.5 AOD data are used with a corresponding

uncertainty of 0.01-0.02 (Holben et al., 2001).

For the analysis of the air mass transport the HYSPLIT model was used (Stein et al., 2015). HYSPLIT model calculates backward and forward trajectories of air masses for simulations of dispersion and deposition at a given location. In order to identify the aerosol sources and to create some statistical basis of the aerosol conditions during DACAPO-PESO, we used ensemble backward trajectories combined with a land cover classification for a temporally and vertically resolved air mass source

attribution TRACE (Radenz and Seifert, 2019). A simplified version of the MODIS land cover (Friedl et al., 2002) is used. As a first step, a 27-member ensemble of 10 day backward trajectories is calculated using HYSPLIT. Meteorological input data for HYSPLIT were downloaded from the Global Data and Assimilation Service (GDAS1; https://www.ready.noaa.gov/gdas1.php) provided by the Air Resources Laboratory (ARL) of the US National Weather Service's National Centers for Environmental Prediction (NCEP). Each ensemble is generated using a small spatial offset in the trajectory endpoint. Whenever a trajectory

is located below the planetary boundary layer (PBL) height provided by the GDAS1 data ("reception height"), the land cover is categorized using custom defined polygons according to land mass boundaries. It is therefore assumed that an air parcel is influenced by the surface type if the corresponding trajectory is below the PBL height. The residence time for each category is then the total time an air parcel fulfilled this criterion by land cover category. This calculation is repeated in steps of 3 h in time and 500 m in height in order to provide a continuous estimate on the source of the air mass and a first indication on the

potential aerosol load.





## 3 Case Studies

In this section, two of the most distinctive cases with lofted aerosol layers that were identified above Punta Arenas are presented in detail. The first one presents a 3-km thick lofted aerosol layer, located in the lowest troposphere between 1 and 4.2 km. Source attribution with TRACE indicated that the layer originated from Central and Central-South Chile, where wildfires occurred at the same time. The second case shows thin (geometrically and optically) aerosol layers, as observed by Polly$^{XT}$, at altitudes of 5.5–7, 9, and 11 km after a long-range transport event from Australia. The retrieved optical properties, indicate that the layers consist of smoke particles. The retrieved optical parameters for the two case studies are summarized in Table 1.

### 3.1 Lofted smoke layers from South America

From 4 February to 5 February 2019, several lofted aerosol layers were observed between 1 and 4.5 km height. Figure 2 shows a two-day overview of main lidar parameters, the Sun-photometer-derived AOD, atmospheric thermodynamics, and a target classification measured by Polly$^{XT}$. As indicated in Figure 2, a free-tropospheric aerosol plume with a geometrical thickness of 3 km was present on 4 February 2019 between 02:00 and 19:30 UTC. The target classification (Baars et al. (2017), Fig. 2d) clearly identifies this aerosol up to about 4.3 km height but was not able to identify the type of the aerosol particles present as a result of weak backscattering signals. The AOD during that day was low, with a mean value of 0.05 at the wavelength of 500 nm. From the beginning of the campaign until March 2019, the mean daily averaged aerosol optical thickness at Punta Arenas was found to be 0.03 (at 500 nm), thus a slightly increased AOD was observed. The aforementioned layer is characterized by low depolarization ratio values that indicate the presence of spherical particles. From the Ångström exponent (Figure 2a), we can conclude that the layer consisted mainly of fine-mode aerosol particles (i.e. particles smaller than $1\,\mu$m).

The following day, on 5 February 2019, a descending layer was observed between 07:00 and 12:00 UTC. The layer extents from 1 to 4.2 km height. In contrast to the layer from the previous day, this one is characterized by higher backscatter and low to moderate depolarization ratio with the most pronounced features found between 1.6 and 2.8 km. The higher values of volume linear depolarization ratio, with respect to the previous day, indicate the presence of non-spherical particles. Similarly to the previous day, the AOD during that event was low, with a maximum and a mean value of 0.09 and 0.05 respectively (at 500 nm), which is however still three times higher than the average. Around 18:00 UTC we observe the formation of a cloud, which according to the target classification consists of ice crystals, thus indicating that ice formation is most likely supported by the advection of the smoke particles acting as INP, as discussed below.

Further context of the air masses arriving at Punta Arenas is given by the HYSPLIT backward trajectory analysis shown in Figs. 3 and 4. The 10-day backward trajectories presented in Fig. 3 show that the vast majority of the air parcels arriving over Punta Arenas at 3000 m height originated from the southern Pacific Ocean. Shortly (1–2 days) before the arrival of the air parcels to Punta Arenas, they passed over the regions of Central and Central-South Chile (Figs. 3, 4a). Fig. 3c demonstrates that no precipitation occurred while the trajectories passed over the land masses. Air masses crossed above lands that were categorized as forest, savanna, grass and barren (Fig. 4b) and belong to South America (Fig. 4a). According to FIRMS (Fire Information for Resource Management System), on the 2nd and 3rd of February 2019, when the air parcels were located





above the aforementioned regions, active fires and wildfires occurred over the region passed by the trajectories (not shown
here). During the summer months (December to February), wildfires are common in Chile, especially between the regions of
Valparaíso and Los Lagos, due to the extremely dry conditions and the prevailing westerlies. Relative humidity in the aerosol
layer (Fig. 3b, Fig. 2e,f) was mostly below 60% and therefore we do not expect any hygroscopicity effects on the aerosol
observed. As the air masses crossed above active fire regions 1-2 days before they were observed by our system, the presence
of partly coated soot particles is expected. Apart from smoke particles, soil dust is also expected to be present in the observed
layer due to turbulent fire-related winds that developed above the burning areas (Nisantzi et al., 2014; Wagner et al., 2018).

The backscatter coefficient (Fig. 5a) shows a similar behaviour for all three wavelengths with mean values of $0.90 \pm 0.25$
$\mathrm{Mm}^{-1}\mathrm{sr}^{-1}$, $0.67 \pm 0.23$ $\mathrm{Mm}^{-1}\mathrm{sr}^{-1}$ and $0.34 \pm 0.11$ $\mathrm{Mm}^{-1}\mathrm{sr}^{-1}$ for 355, 532 and 1064 nm respectively. The maximum
values for all wavelengths are observed around 1.5 km. Backscatter-related Ångström exponents of $0.76 \pm 0.23$ and $0.97 \pm$
$0.29$ at the wavelength pairs of 355/532 nm and 532/1064 nm, respectively were determined by lidar observations (Fig. 5c),
and indicate the presence of coarse particles within the smoke layer. The extinction coefficient (not shown here) reached values
up to 65 $\mathrm{Mm}^{-1}$ at 355 nm and 39 $\mathrm{Mm}^{-1}$ at 532 nm within the smoke layer. These values are typical for smoke layers and
within the range of values previously reported in the literature (Amiridis et al., 2009; Kanitz et al., 2014; Haarig et al., 2018).
Given the depolarization ratio values (Fig. 5d and discussion below), the coarse particles are dust particles (Tesche et al.,
2009a; Ansmann et al., 2009). In comparison, Foth et al. (2019) reported a backscatter-related Ångström exponent (532/1064
nm) of $0.56 \pm 0.21$ for a Patagonian dust layer and of $0.61 \pm 0.1$ for a smoke plume originating from Australia. Below 2.4 km
the spectral dependency of the light absorbing smoke particles is evident. Above that altitude, backscatter-related Ångström
exponents are roughly the same at both wavelength pairs. Mean lidar ratio (Fig. 5b) values were $49 \pm 12$ sr and $24 \pm 18$ sr for
355 and 532 nm, respectively. The lidar ratios reported here are in consistency with previously published results (Balis et al.,
2003; Müller et al., 2005; Tesche et al., 2009b; Giannakaki et al., 2010; Tesche et al., 2011; Baars et al., 2012; Kanitz et al.,
2013; Baars et al., 2016). However, it should be noted that differences in the burned vegetation type, location (e.g. Eastern
Europe, Siberia, Canada), fire type (smoldering or flaming combustion) and the age of the smoke particles are inducing the
relatively large range of values reported in the literature. Within the smoke plume, linear particle depolarization ratio (Fig.
5d) ranges from 2% to 14% (mean: $7.6 \pm 3.6\%$) at 532 nm with a maximum being observed between 2.2 and 2.3 km. The
slightly elevated particle linear depolarization ratio at this altitude can be explained by the presence of irregularly shaped soil
dust particles (Nisantzi et al., 2014) along with some partly coated soot particles. The smoke-dust mixture is mostly confined
between 2.0 and 2.8 km and it was well identified by the target classification algorithm as a partly non-spherical mixture (Fig.
2d).

The potential impact of this event on cloud and precipitation evolution and life cycle can be investigated by means of CCN
and INP concentrations. CCN and INP number concentrations have been derived for this plume (1- 4.2 km), uncertainty
range for this approach is of a factor of 2 and 3, respectively (Mamouri and Ansmann, 2016). The lidar-derived total CCN
concentration, given for 0.2% supersaturation with respect to liquid water, has a layer mean value and a standard deviation
of $642 \pm 192$ $\mathrm{cm}^{-3}$. The contribution of dust to the total CCN concentration is about 6% and the rest is attributed to soot
particles. The obtained CCN number concentration is found to be strongly increased compared to other values reported for this





region. Hamilton et al. (2014) reported much lower monthly mean values of CCN concentration for January (50 to 100 $\mathrm{cm}^{-3}$).

For comparison, Schmale et al. (2018) reported mean CCN concentrations within the range of 1200 to 1500 $\mathrm{cm}^{-3}$ (at 0.2% water supersaturation) for central Europe. We can therefore conclude that the CCN concentration was significantly increased due to the advection of smoke/dust, but was still below the levels typically observed in Central Europe. The parameterizations used for the derivation of the INP number concentration (Ullrich et al., 2017) correspond to soot particles in the immersion freezing temperature regime (see Fig. 2f). From the lidar observations, a layer mean of 284 ± 166 INP per L (at -20 °C)

was derived. This value corresponds only to soot particles since the contribution of dust particles to the total INP number concentration at the chosen temperature is weak and almost negligible. The INP number concentration is found to be elevated for the Southern Hemisphere, especially when compared with the typical values of the region (annual 1-5 $\times 10^{-3}$ INP/L for an activation temperature of -15 °C), where marine aerosol is the dominant aerosol type of INPs (Vergara-Temprado et al., 2017).

### 3.2 Long-range transported smoke layers from Australia

The next event of lofted aerosol layers that occurred during the DACAPO-PESO campaign was observed about one month after the previous layers (sect. 3.1), on 11 March 2019. The temporal evolution of the event, by means of basic lidar parameters, aerosol target classification, AOD, and thermodynamic profiles, is depicted in Figure 6. The AOD during that day was low, with a daily mean of 0.03 (500 nm). The layers were observed between 02:00 and 13:00 UTC and between 5 and 12 km altitude. In contrast with the observations from 5 February 2019, where the layer was located between 1 and 4.2 km height, these layers

were geometrically thin (approx. 1 km), and are located at considerably higher altitudes (see Fig. 6b, 6d) but stacked upon each other. The backscatter coefficient values vary within the three layers, with the maximum being observed for the highest layer, which was observed at around 11-12 km height (Fig. 6b). The highest layer can be separated from the other layers in terms of depolarization ratio as it is the only one that consists of particles causing depolarization (Fig. 6c). Due to the low particle concentration and the uncertainties induced by the high altitude of the layers (decreasing signal-to-noise ratio with height), the

target classification algorithm could not assign a specific aerosol type to the particles observed within the layer (Fig. 6d). Apart from these high altitude layers, we also observed clouds, mostly confined in the lowest 2 km. Thus in this specific case, we have no evidence that the aerosol did influence cloud formation, but nevertheless the particles may have the potential to do so if atmospheric conditions change.

This observed stratification of the multiple aerosol layers is a characteristic feature of plumes advected over transcontinental

scales (Müller et al., 2005). The long-range transport can be confirmed based on the back-trajectory analyses that are presented in Figs. 7 and 8. As indicated by the 10-day HYSPLIT backward trajectories (Figs. 7, 8a), all the air parcels arriving at Punta Arenas on 11 March 2019, at 06:00 UTC and at 11 km altitude, originate from the southwestern coast of Australia and travelled at high altitudes above 10 km (Fig. 7a). Before crossing the South Pacific Ocean, the air masses (above 5.5 km) appear to have a significant residence time above lands characterized as savanna/shrub and slightly less time above regions of grass, crops and

forests (Fig. 8b). According to FIRMS, when the air masses where above the southern parts of Australia, Tasmania and New Zealand (approx. 9 days before they were observed above Punta Arenas), there was a high count of active fires (not shown here) in all the aforementioned regions. Similarly to the previous case, there are no hygroscopicity effects on the particles within





the observed layers, since the relative humidity (Fig. 7b) of the air masses observed above Punta Arena was less than 25%. In addition, there was no washout of the carried particles via wet deposition since the modeled total precipitation was almost
negligible (less than 1 mm, Fig. 7c).

The height profiles of the backscatter coefficient, Ångström exponents and particle linear depolarization ratio as derived from the lidar measurements of 11 March 2019, between 2:45 and 3:40 UTC for the three lofted layers are shown in Fig. 9. Profiles of extinction coefficient and lidar ratio are not presented due to low signal to noise ratio at the altitudes of interest.

The three different layers are characterized by pronounced backscatter coefficients (Fig. 9a). More specifically, the backscat-
ter coefficient reached values up to 0.15 and 0.08 $\mathrm{Mm^{-1}sr^{-1}}$ for the layer located between 5.5 and 7 km, 0.09 and 0.06 $\mathrm{Mm^{-1}sr^{-1}}$ for the middle layer located around 9 km, and 0.35 and 0.18 $\mathrm{Mm^{-1}sr^{-1}}$ for the highest smoke layer observed around the altitude of 11 km (at 532 and 1064 nm respectively). The backscatter-related Ångström exponent (Fig. 9b) at the wavelength pair of 532/1064 nm has a similar behavior for the lowest and highest layers with values up to 0.8, suggesting wavelength dependency and indicating the absence of coarse-mode aerosol particles. For the layer located at the altitude of 9
km, the Ångström exponent is considerably lower, with a mean of $0.22 \pm 0.07$, suggesting the presence of larger particles.

The particle linear depolarization ratio (Fig. 9c) is height-independent up to 10 km with values of less than 2%. Similar values of particle linear depolarization ratio (< 3%) have been reported by Baars et al. (2011) for aged smoke in the Amazon rain forest during the dry season. Values of that range indicate that most of the particles observed are spherical in shape (i.e. smoke particles). More specifically, for the middle layer with the very low Ångström exponent values, we can conclude that
most likely it consists of rather large, spherical particles of continental aerosol. Above 10 km height, we observe a sharp increase of the particle depolarization ratio, until it reaches its maximum (9.5%) within the highest aerosol layer, at an altitude of 11 km. Mean depolarization ratio values within this layer were $8.3 \pm 1.02\%$ at 532 nm. This increase of the depolarization ratio reveals that within the highest and driest (RH< 10%) layer there were irregularly shaped, partly coated soot particles. Dry, depolarizing smoke layers in the upper troposphere/lower stratosphere have been observed previously. Burton et al. (2012)
observed over Mexico City, a similar smoke layer at a height of 8 km with a depolarization ratio of 9% at 532 nm. The increased depolarization ratio for dry smoke layers in the stratosphere was observed by Haarig et al. (2018) and compared to more humid smoke layers in the lower troposphere with low values for the depolarization ratio (<5%). For an extensive literature overview on aged and fresh smoke properties based on multiwavelength lidar observations, readers may refer to Haarig et al. (2018).

INP number concentration was derived for the upper layer, which was observed around the altitude of 11 km. Since the temperature in the layer is below -35°C (Fig. 6e), the deposition nucleation freezing parameterization for soot of Ullrich et al. (2017) is used. The whole backscatter coefficient was attributed to smoke particles, as the separation of dust and smoke does not work for smoke layers in the upper troposphere and lower stratosphere with their enhanced depolarization ratios. For an assumed supersaturation with respect to ice of 20%, an INP number concentration of $1.07 \times 10^3 \pm 714.9$ $\mathrm{L^{-1}}$ was derived for
ambient temperature. For the same supersaturation and an assumed fixed temperature of -55°C, the INP number concentration would be higher $1.96 \times 10^3 \pm 759.5$ $\mathrm{L^{-1}}$. Immersion freezing and CCN activation are not important at that temperature range. Based on satellite observations, a supersaturation with respect to ice of 15-20% is found to be the best choice in cirrus clouds in



the northern midlattitudes (Lamquin et al., 2012). The high numbers of smoke INP observed at 11 km height will impact cirrus formation as soon as cooling or lifting of the air mass leads to an increase of relative humidity. Homogeneous ice formation will

be suppressed, as it requires supersaturations of around 40% with respect to ice (KÄrcher and Jensen, 2017), while sufficient INP are present to form ice at much lower humidities.

## 4 Long-term analysis of occurrence of lofted layers

Similarly to the ALPACA campaign (Kanitz et al., 2013; Foth et al., 2019), lofted layers were observed more frequently than expected in the pristine environment of Punta Arenas in 2019. Figure 10 gives an overview of the occurrence of lofted

layers observed by the lidar in Punta Arenas during the DACAPO-PESO campaign until the end of 2019. In total, lofted layers were observed on 16 different days (3.9%). Most of the layers were both optically and geometrically thin. According to AERONET, the averaged daily mean AOD of the days, on which lofted aerosol layers occurred, was as low as 0.05 at 500 nm wavelength. In addition, almost all layers were non-depolarizing, indicating that they consisted mainly of spherical particles. The only exceptions were the layer presented in Sec. 3.1, and a thin layer that was observed on 3 December 2019

(slighlty depolarizing). Nevertheless, such layers have the potential to influence cloud formation in this environment which is supposed to be almost pristine. Even small perturbations could therefore influence cloud properties strongly, while in the polluted northern hemisphere such perturbations would not be observable. This notion is also corroborated by the study of Villanueva et al. (2020), who discuss the hemispheric differences in the effect of mineral dust on heterogeneous ice formation.

10-day HYSPLIT backward trajectories for all the layers were performed, and revealed that the main origins of the air

masses were the southern Oceans including Australia and New Zealand. To be more specific, layers observed until October 2019 originated mainly from the Southern Pacific and the Southern Ocean (and consisted of marine particles mostly), with only a fraction tracing back to the southern tips of Australia and South America (e.g. Sec. 3.1). Their origin and optical properties (not presented here), indicate that they were likely mainly consisting of large and spherical particles. The layers observed in November and December 2019 were originating mainly from Australia. Given the origin of the air masses, their altitude and

geometrical as well as optical properties, the layers can be directly linked to the outstanding Australian bushfires of 2019-20. For a more detailed overview of these Australian bushfires on stratopsheric aerosol conditions refer to Ohneiser et al. (2020).

## 5 Conclusions and outlook

During the first 14 months of the DACAPO-PESO campaign, significant lofted layers were observed regularly above Punta Arenas. Two events were studied and presented in detail, based on observations from the multiwavelength Raman lidar Polly$^{XT}$.

The Raman lidar observations in combination with the HYSPLIT backward trajectory analysis allowed the identification of the aerosol. In the first case, the lofted aerosol layers originated from Central and Central-South Chile and were identified as smoke layers, mixed with soil dust. Smoke and dust particles were probably mixed near the source and lofted to higher altitudes via



convection. In the second case, several geometrically and optically thin smoke layers were observed, after long-range transport from Australia.

The differences in the obtained optical properties between the observed layers, reflect differences in the amount and the age (chemical composition) of the observed smoke particles. As it can been seen from Table 1, the 532-nm particle depolarization ratio for the smoke and soil dust mixtures is slightly lower than the one for the pure smoke layer. In the first case, the smoke particles within the layer were freshly emitted (emitted 1-2 days before observation), while in the second case, the smoke particles were aged (observed 9 days after emission). Previous studies have shown that the depolarization ratio values at 532

nm for smoke freshly emitted into the atmosphere are generally low (due to the sphericity of the newly formed particles) and even lower for aged smoke particles (Burton et al. (2012, 2015); Haarig et al. (2018)). The elevated depolarization values of the smoke layer of the first case can additionally be attributed to the presence of soil dust and the resulting elevated mixing ratio of smoke and dust particles (Sugimoto and Lee, 2006; Tesche et al., 2009a). In addition, differences in RH are one of the main reasons for the different depolarization ratios. The backscatter-related Ångström exponent at the wavelength pair of

532/1064 nm exhibit similar behavior for both layers, with slightly higher values for the aged pure smoke layer, indicating a predominance of fine mode particles.

In comparison to the prevailing marine conditions in the region of Punta Arenas, biomass burning sources acted as an effective source of CCN and INP. Smoke advection from Central and Central-South Chile affected significantly the available CCN and INP in the lower troposphere. Both CCN and INP number concentration were higher than usual for the first case and

likely contributed to the characteristics of an ice cloud that was observed only a couple of hours later (cloud formation, Fig. 2d). In addition, smoke particles advected from Australia at high altitudes were found to likely facilitate the formation of ice crystals.

The Southern Oceans and more specifically Punta Arenas, are usually characterized as pristine environments, where clean marine aerosol conditions prevail. However, this study demonstrates that this is not always the case. Lofted aerosol layers,

transported either from short or long distances, occur regularly and influence radiation and cloud formation processes above the region. Even small perturbations influence cloud properties strongly above Punta Arenas, while in the polluted northern hemisphere such perturbations would not be observable.

*Data availability.* The Polly$^{XT}$ lidar data are available at TROPOS upon request (polly@tropos.de). GDAS1 meteorological data are available and can be downloaded at the webpage of NOAA (https://ready.arl.noaa.gov/READYamet.php). AERONET sun-photometer data were

downloaded from the AERONET web page (AERONET, 2019). Trajectories are calculated with the NOAA HYSPLIT model (HYSPLIT, 2019).

*Author contributions.* MR, PS, CJ, RE, BB and FZ are responsible for the instrumentation and collected the observational data. AAF, ZY and HB analyzed the lidar and AERONET data. MR analysed the HYSPLIT data and provided the corresponding figures. MH derived the INP





and CCN concentrations. UW has contributed in several discussions regarding the optical effects of smoke. AAF prepared the manuscript in
close cooperation with HB and AA.

*Competing interests.* The authors declare that they have no conflict of interest.

*Acknowledgements.* Parts of the research leading to these results have received funding from the European Union's Horizon 2020 research
and innovation programme under grant agreement no. 654109 (ACTRIS).



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



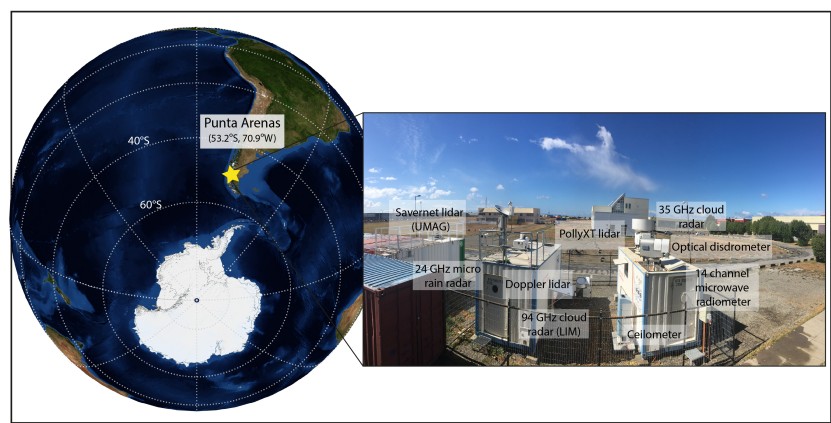

**Figure 1.** Map of the location of Punta Arenas (indicated by a yellow star) in Chile, South America. The LACROS observational facility, as deployed at UMAG, can be seen on the right side.

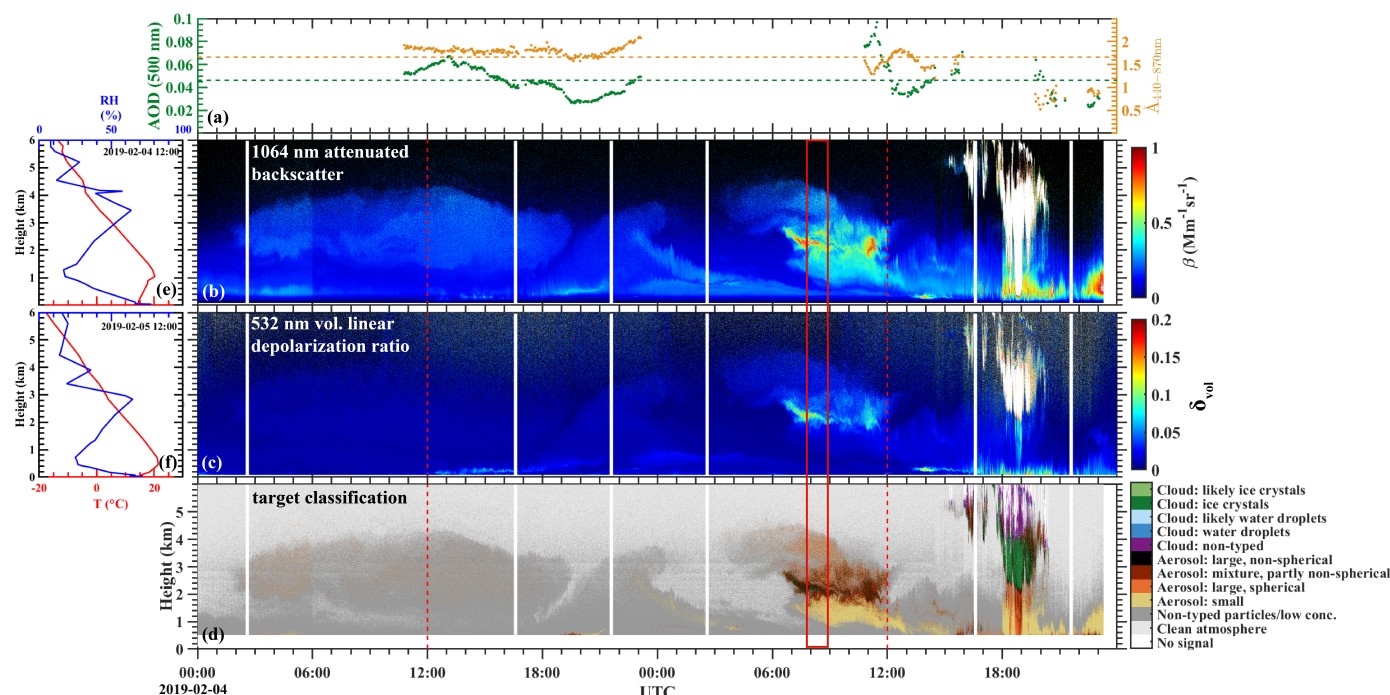

**Figure 2.** From the upper to the lower panel: (a) AOD at 500 nm and Ångström exponent for 440/870 nm (dashed lines indicate the mean), (b) attenuated backscatter coefficient at 1064 nm ($Mm^{-1}sr^{-1}$), (c) volume depolarization ratio (% at 532 mn), and (d) target classification reflecting the atmospheric conditions over Punta Arenas from 4 February to 5 February 2019. Radiosonde launches are depicted with red dashed lines. The corresponding temperature and relative humidity profiles can be seen on the left panels (e, f). Vertical white lines indicate the automatic calibration of the linear depolarization ratio (Engelmann et al. (2016)). The averaging period for the profiles shown in Fig. 5 is indicated by the red rectangular.

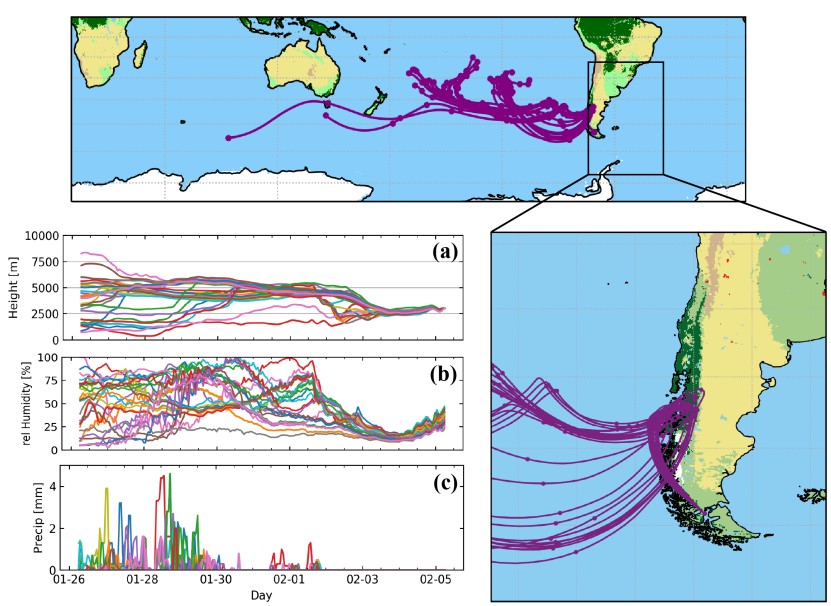

**Figure 3.** Top: the 10-day HYSPLIT backward trajectories (http://www.arl.noaa.gov/ready/hysplit4.html) ending at Punta Arenas, Chile, on 05 February 2019, 06:00 UTC, at 3 km altitude. Panels below show (a) the altitude (m) at which the air masses were traveling, (b) the relative humidity (%) of the air masses and (c) the precipitation in mm. Right: Zoom of top panel.



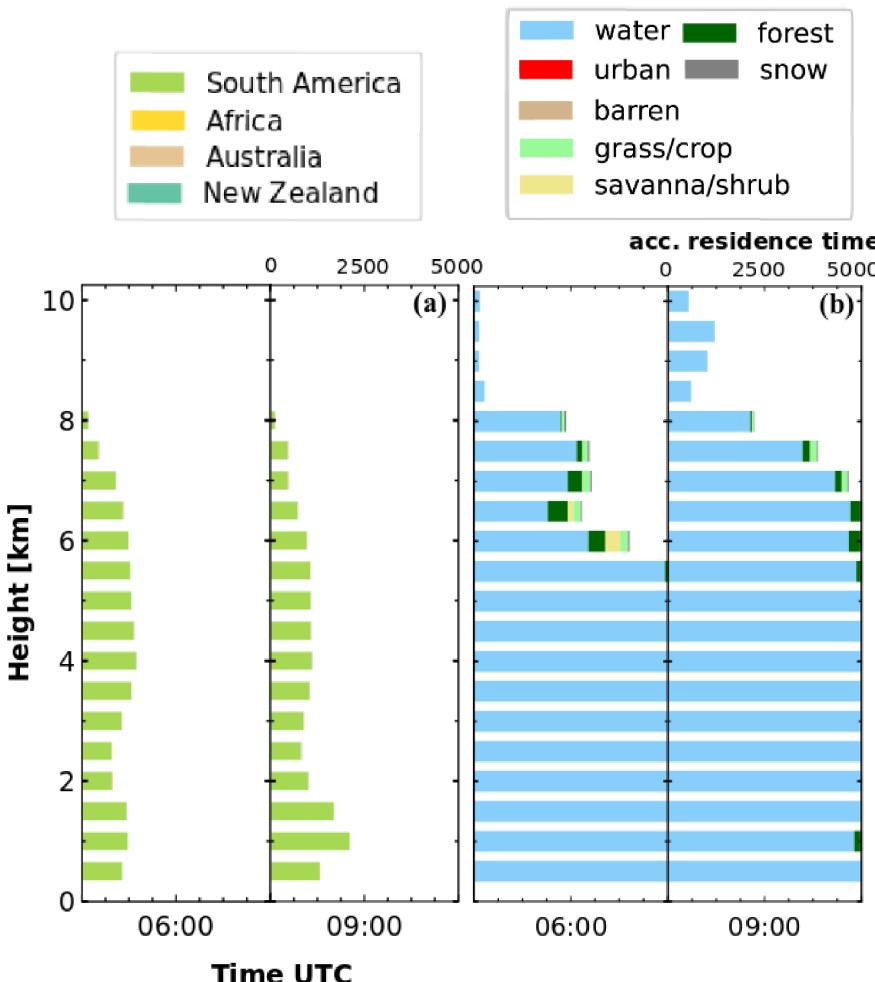

**Figure 4.** Accumulated residence time (h) of the air masses arriving at several heights over Punta Arenas on 05 February 2019. Above (a) geographical regions and (b) type of surface.

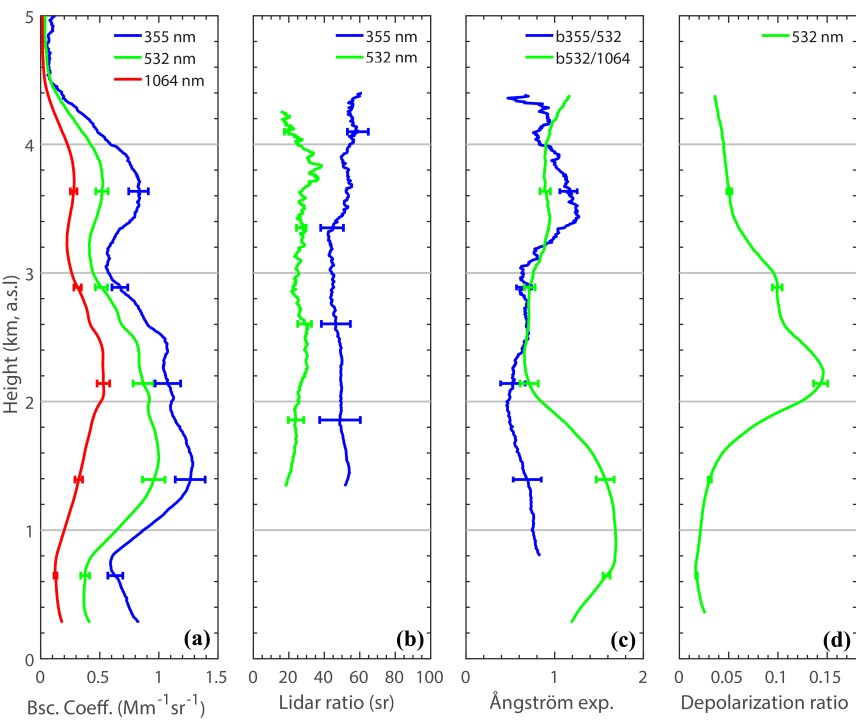

**Figure 5.** (a) Particle backscatter coefficients (at three wavelengths), (b) the extinction-to-backscatter ratios, (c) the corresponding backscatter-related Ångström exponents, and (d) particle linear depolarization ratio measured at Punta Arenas on 05 February 2019, 07:51–08:57 UTC.





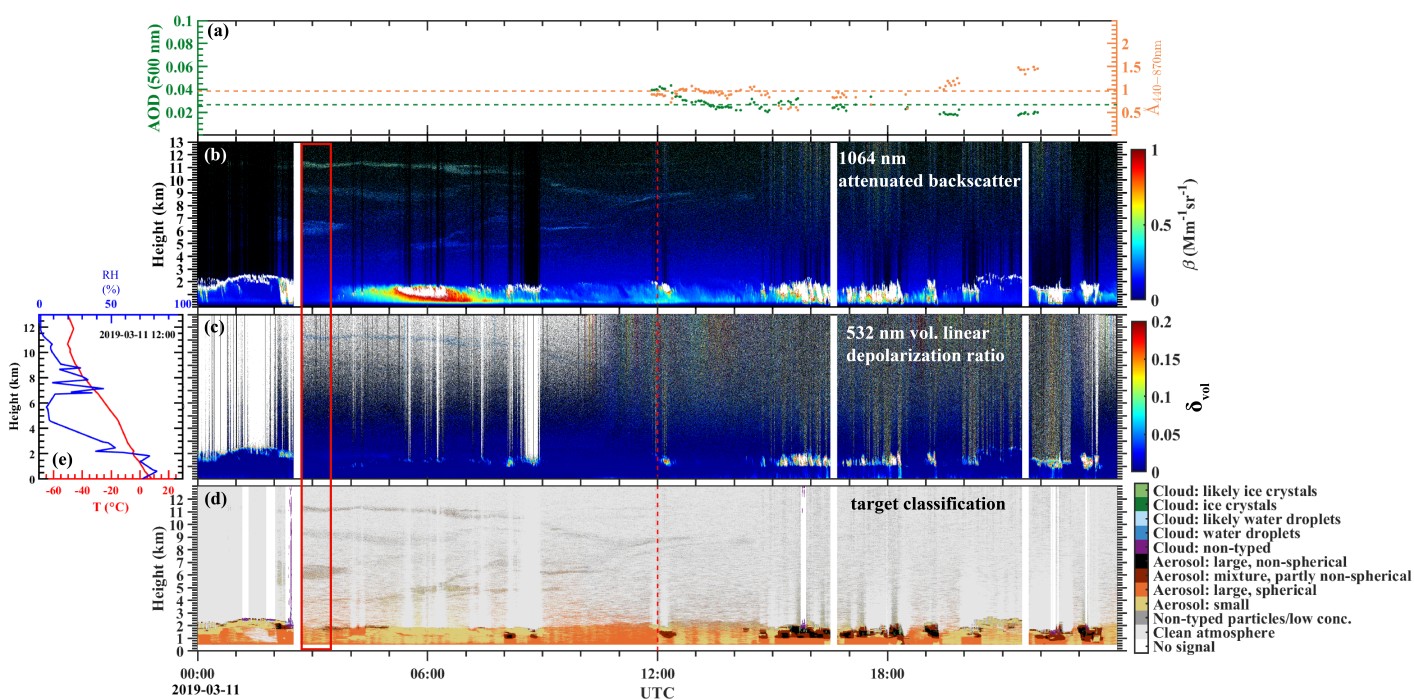

**Figure 6.** Same as Fig. 1 but for 11 March 2019. The averaging period for the profiles shown in Fig. 9 is indicated by the red rectangular.

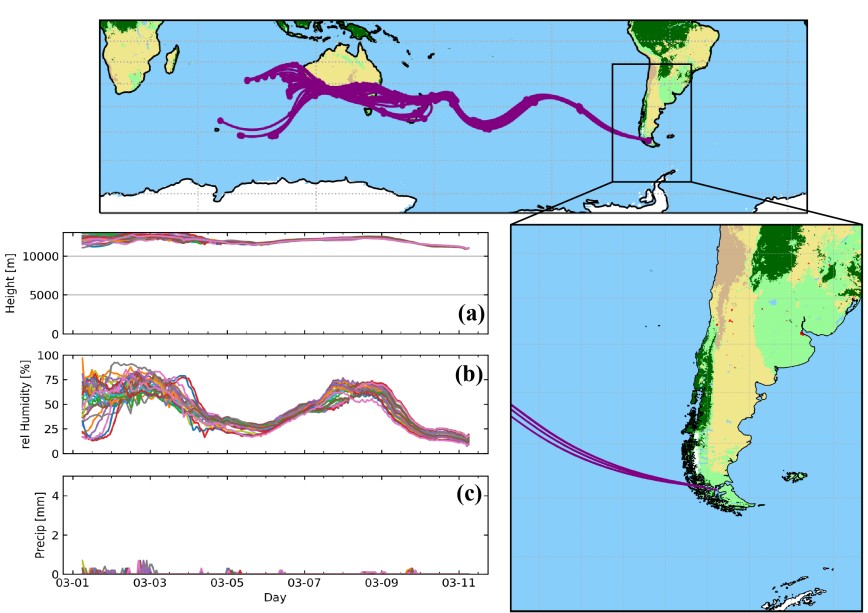

**Figure 7.** Same as Fig. 2 but for 11 March 2019, 06:00 UTC, at 11 km altitude.





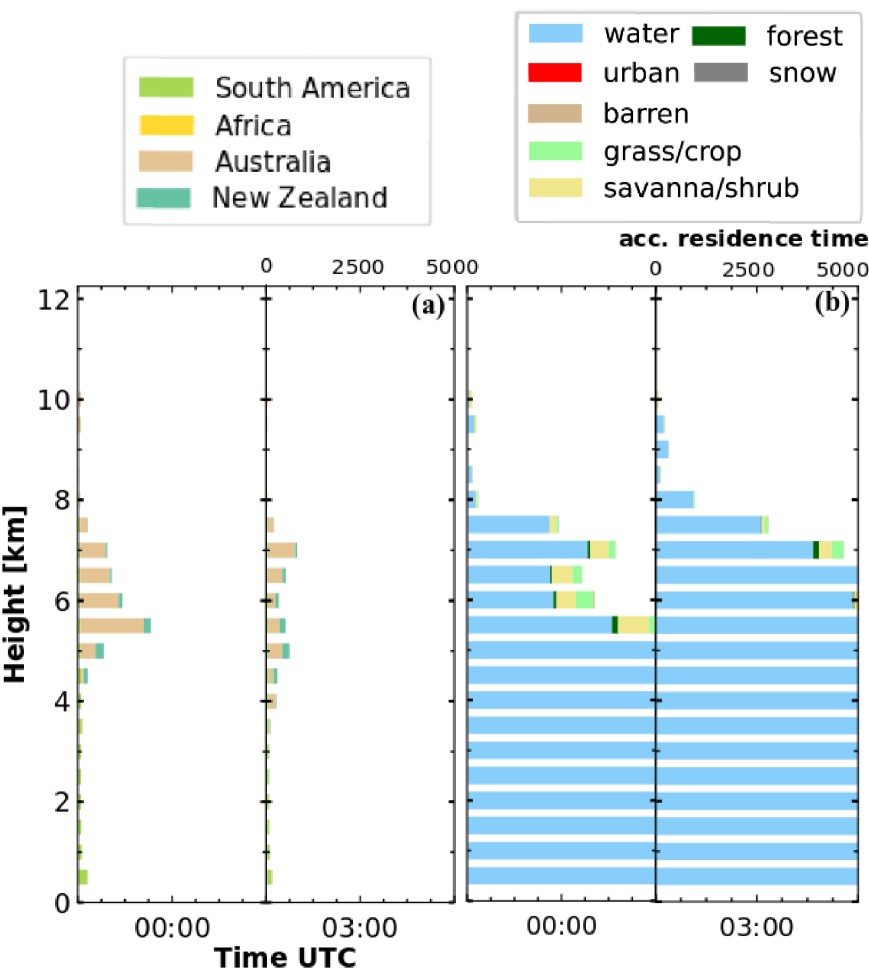

**Figure 8.** Same as Fig. 3 but for 11 March 2019.



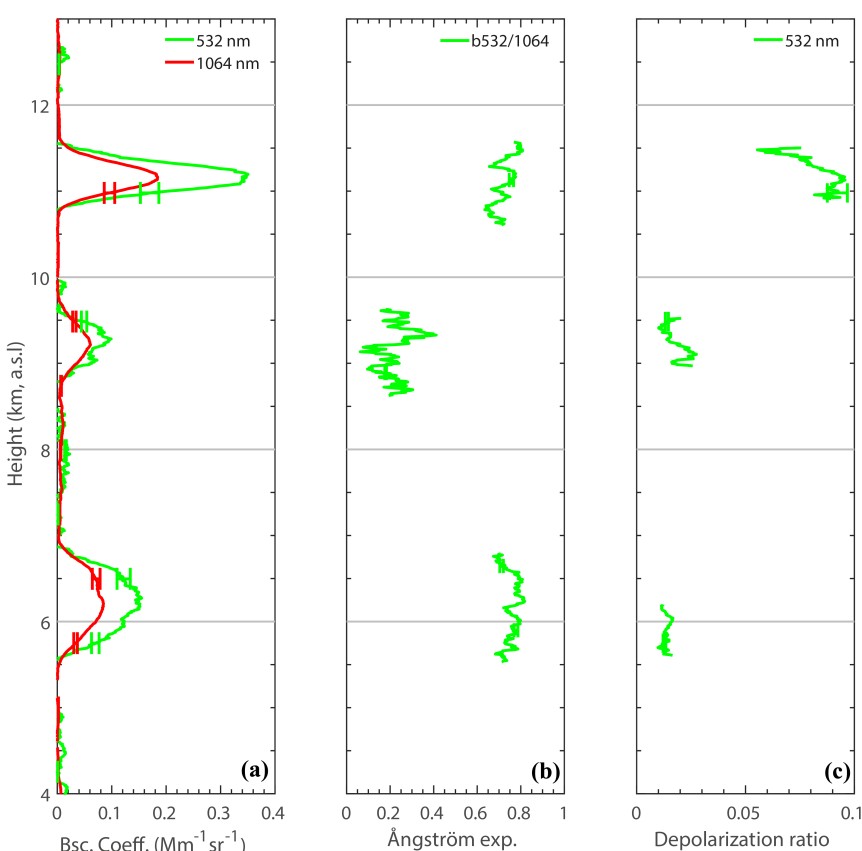

**Figure 9.** Mean profiles of (a) particle backscatter coefficients (at 532 and 1064 nm), (b) the corresponding backscatter-related Ångström exponent (for 532/1064 nm), and (c) particle linear depolarization ratio at 532 nm measured at Punta Arenas on 11 March 2019, 02:45- 03:40 UTC.



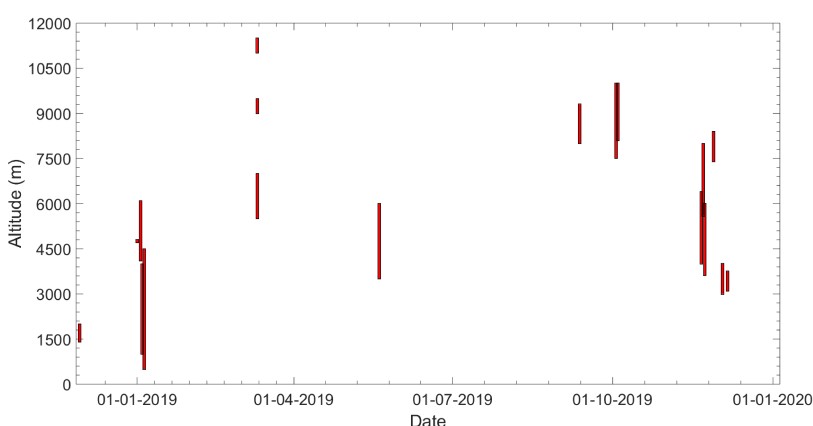

**Figure 10.** Occurrence and height boundaries of lofted aerosol layers in Punta Arenas between 29 Nov. and 31 Dec. 2019.





**Table 1.** Overview of aerosol optical properties as derived from Polly$^{XT}$ LACROS in Punta Arenas on 05 February 2019 and 11 March 2019 (values correspond to the layer located at 11 km). $\beta$ is the backscatter coefficient, S the lidar ratio, $\delta$ the particle linear depolarization ratio and Å the backscatter-related Ångström exponent. The subscript indicates the corresponding wavelength. All values correspond to averaged (mean) values per examined layer along with the standard deviation.

| Date | $\beta_{355}, \beta_{532}, \beta_{1064}$ $(Mm^{-1}sr^{-1})$ | $S_{355}, S_{532}$ $(sr)$ | $\delta_{532}$ $(\%)$ | $Å_{355/532}, Å_{532/1064}$ |
|---|---|---|---|---|
| 5 February 2019 | $0.90 \pm 0.25$ | $49 \pm 12$ | $7.6 \pm 0.23$ | $0.76 \pm 0.23$ |
|  | $0.67 \pm 0.23$ | $24 \pm 18$ |  | $0.97 \pm 0.29$ |
|  | $0.34 \pm 0.11$ |  |  |  |
| 11 March 2019 | - | - | $8.3 \pm 1.0$ | - |
|  | $0.20 \pm 0.11$ | - |  | $0.72 \pm 0.04$ |
|  | $0.10 \pm 0.05$ |  |  |  |