# Peer review of "Biomass burning aerosols in the southern hemispheric midlatitudes as observed with a multiwavelength polarization Raman lidar"

_Atmospheric Chemistry and Physics, 2020_

## Referee Comment (RC1) · Anonymous Referee #1 · 24 Jun 2020

This paper presents two aerosol transport events over Punta Arenas observed by a 3-wavelength lidar. Aerosol transport in the middle and upper troposphere is a hot topic to improve our understanding of the climate system. Nevertheless, the study carried out in this article is rather sporadic and does not reflect the title chosen, which suggests scientific work over a large geographical area, important for studies of aerosol-cloud interactions. One should be clearer and write in the title that it is above Punta Arenas. The scientific information content of this article seems to me too light for publication at ACP. It should be given more relevance and better explain certain delicate points of the approach.

[Figure]

To give more content to this article, using the existing lidar dataset, it would be better to contextualize the study and to use complementary observations. These observations could be those of spaceborne measurements and one can think primarily to the CALIPSO and MODIS missions. Thus, several years of observations would be available, which would make it possible to have a more statistical study on a larger scale effectively covering the southern hemispheric midlatitudes.

Different emission, injection and transport conditions have been described in the scientific literature with the help of tools such as lidar instrumentation associated with the study of air mass back trajectories. It would have been interesting to have a more representative synthesis in the introduction of this article which could be based on the major international field campaigns. These different campaigns are also particularly good examples of more global approaches.

The algorithmic approach used requires a sufficient amount of aerosols in the atmospheric column. In particular, a sufficiently high aerosol extinction coefficient is required. For the values presented in this paper, the uncertainties are extremely high and make it difficult to make the data representative for conclusions. A simple error calculation already shows that the Ångström exponent are associated with a very high uncertainty which reduces their power of discrimination of aerosol types. It is essential to discuss all sources of uncertainty when analysing extremely low aerosol signature on lidar profiles. This discussion is totally absent from the paper, which significantly reduces its scientific value. There are no error bar on the profiles presented and they should be added in parallel with the discussion of uncertainties.

Points of more or less importance in the text (this list is not exhaustive as the article must undergo major revisions before it can be reviewed again).

Abstract

Define CCN and INP.

[Figure]

Section 2.1

Why present all the instruments when only PollyXT is used?

What justifies the size of the sliding windows? A sensitivity study should be presented. The cut-off frequencies of the low-pass filters used should be known. This type of filters also have bounces that can lead to signal distortion in low signal conditions.

Formulas (1) and (2) are not clearly explained. Each variable needs to be defined and the numerical values chosen must be better justified. The sentence " To reduce the uncertainties, a specific parameterization for smoke is under development." doesn't have much relevance if it's not made for the paper.

Section 2.2

In level 1.5 data, clouds may be present, especially high clouds of type Ci. This may be a difficulty, especially for the second case study.

The way in which back trajectory portions are selected, based on a single planetary boundary layer criterion may not be sufficient, especially for biomass fires where the injection heights can be well above the top of the planetary boundary layer.

Section 3

In Table 1, the altitude of the aerosol layers should be indicated.

Give the uncertainties on extremely low AOD. The very small aerosol extinction coefficients limit the interest of calculating an Ångström coefficient, it must be better justified using error calculations.

For the identification of the fires that should be shown, the fire product of MODIS could help.

The references given for the aerosol extinction coefficient of biomass burning aerosols to justify the low values are not sufficient. There are much higher values in the biomass

burning aerosol layers.

Explain why this is evident in the sentence " Below 2.4 km the spectral dependency of the light absorbing smoke particles is evident.".

How is supersaturation identified?

How is the 6% attributed to soot particles justified?

The role of soot as INPs is difficult to demonstrate in this study. If soot aerosols are young and aspheric, they may well serve as icy nuclei. The microphysical structure must be close to that of ice. In the presence of a coating, that doesn't work. Here, the aerosols are old and therefore have a coating.

For the second case study, what proves that there is no contamination by fine cirrus?

What is the purpose of the sentence " Thus in this specific case, we have no evidence that the aerosol did influence cloud formation, but nevertheless the particles may have the potential to do so if atmospheric conditions change.", knowing that the identified aerosols are at a much higher altitude than the clouds at 2 km AMSL?

In this second case study, it would also be good to show the fires.

In this case, the Ångström exponent seems calculated on the aerosol backscatter coefficient, one must be aware that it is often different from the one calculated on the aerosol extinction coefficient.

Section 4 It needs to be supplemented using interannual satellite observations.
* * *

---

## Author Comment (AC1) · 12 Jul 2020

The authors would like to thank the reviewer for carefully reading the manuscript and providing us with valuable feedback.

With this response, we intent to provide some information regarding some points that the reviewer made.

First of all, we agree that the title chosen for the manuscript was unintentionally misleading. Thanks for pointing that out. The title will be changed accordingly, in order to better reflect the contents of the manuscript, which focuses on the ground-based lidar

observations made during a long-term (1.5 years) field campaign in Punta Arenas. A possible new title could be "Advection of biomass burning aerosols towards the southern hemispheric mid-latitude station of Punta Arenas as observed with multiwavelength polarization Raman lidar".

Since we do not intent to cover the southern hemispheric midlatitudes as a whole, we will not include any complementary satellite observations which would be far beyond the scope of this paper. In this paper, we want to address the smoke advection towards the usually pristine environment of Punta Arenas.

A more detailed response to the points raised will follow on a later stage.

---

## Referee Comment (RC2) · Anonymous Referee #2 · 19 Jul 2020

This work presents the description with lidar technique of two case studies of lofted aerosol layers over an interesting location in the South Hemisphere, for its importance as observation point for southern mid-latitudes. The layers have been identified according to transport path and possible sources and described in terms of aerosol optical properties and potential for cloud condensation and ice nuclei. A short overview of the lofted aerosol layers observed for 1 year is also included. Although the relevance of the addressed topic can be guessed, the work does not present enough scientific quality to be included in ACP. My main criticisms are the following ones (not going into detail of many particular examples throughout the manuscript):

[Figure]

Scientific significance I find that the approach and results do not represent a substantial contribution in the scope of the journal. The measurement and study of single cases can be of interest in the case that they are exceptional because of their nature, intensity, etc., or if a really deep, detailed description and analysis is given (including proper literature comparisons, etc.). The cases presented here do not fulfill those criteria, and considering that the authors have a longer database of the site, with 16 cases already identified, it would have been more appropriate if a full statistical study of the period had been performed, for example.

Scientific quality- "Introduction" The introduction is poor in several aspects. The justification of the importance of studying biomass burning aerosols, which variables are we interested and why, the relevance of calculating CCN and INP concentrations, etc., is completely missing. There is also no mention to the specific and clear objective of the work and the approach, and which is its impact and novelty. Also, the state of the art should be more complete with references on aerosol (maybe not only lidar) studies at those latitudes, or of the same aerosol type, etc. Some references from the existing networks (only mentioned in the following section without references) should also be discussed.

Scientific quality- "Experiment and Instrumentation" First, the title should also include "methodology". The description of the lidar products does not include any mention to the uncertainties. For the analyzed cases, this is of crucial importance, because of the weakness of the detected layers. There is no explicit mention to several quantities that are later discussed in the manuscript, as AE and lidar ratio. The quality assurance is mentioned but only in a sentence that has no link with the rest of the paragraph. The part about CCN and INP calculation is barely explained, with several symbols not defined. There is only one sentence about sun-photometer, without mention to AERONET or why the level 1.5 is chosen. When writing about "TRACE", the authors should have given some information on how the graphs 4 and 8 are created. Are these accumulated times referring of all the 27 trajectories of the ensemble? In this section,

the target classification scheme is completely missing, as it is also some mention to the radiosondes or the fire detection algorithms as ancillary information. The method for calculating geometrical thickness of the layers is also not specified.

Scientific quality- Results There are several sentences that are too strong with so weak explanations and discussions. For example, "The higher values of volume linear depolarization ratio, with respect to the previous day, indicate the presence of non-spherical particles". Figures 3 and 4 are not clear enough for the unclear description given in the text (maybe, if there was different colour for the trajectory section that is over the land you could really distinguish different periods). The description of optical properties is unclear and sometimes not correct in the view of the graph, and there are several statements claimed to be true or evident without any explanation, discussion or reference. In general, the analysis is not discussed and contrasted with enough literature. Therefore, there are some crucial issues that are very poorly mentioned (e.g., the lidar ratio values are strongly dependent on the vegetation, combustion types, ageing, etc. It is stated in the manuscript, but no examples of literature are given, and no connection with the presented case). There are some sentences in this section that should be in the methodology (e.g. line 180). There are missing references and discussion also in the values and interpretation of INP. In the second study case, the 355 nm profile is missing with no explanation. This case has also the same weaknesses as for the previous one. The long-term analysis is very vague and does not give enough information. This section should have been the main part of the manuscript, with a complete analysis of all the cases and some statistics. There is no explanation on how the layers were identified, etc.

Scientific quality- Conclusions There are several discussions and references to literature that should not appear (explicitly) here, and that actually were not included in the results section. There are also statements in the conclusions that have not been shown in the results (e.g. lines 305-306). Presentation quality The general quality of the text is not good enough. There are many mixed tenses (future tense should not be

used, and sometimes past and present are mixed in the same sentence referring to the same thing). The information is in general not well structured. Some of the paragraphs should have been divided into 2-3 (e.g. lines 66-82). The ideas are often in wrong order, there are many sentences out of place (e.g. lines 150-151 or 163).

In conclusion, I find that a deeper analysis should be done to present this work. Moreover, the manuscript must be re-thought and re-written to avoid the major previously mentioned problems.

---

## Author Comment (AC2) · 30 Jul 2020

The authors would like to thank the reviewers for their effort in reading the manuscript and providing constructive feedback.

In this study, we tried to demonstrate that the pristine environment of Punta Arenas is regularly disturbed by events of smoke aerosol advection from either short or long distances. In most studies, the Southern Oceanic regions are depicted as pristine environments, where marine conditions dominate. The observations from the DACAPO-PESO campaign, however, revealed that regularly the troposphere above Punta Arenas is far from pristine but influenced by an increased aerosol burden. Obviously, this message

could not be transferred to the reviewers. To our knowledge there are not many studies investigating the aerosol burden in the troposphere above Southern Hemispheric mid-latitude locations (often referred to as Southern oceans). We therefore think, it would have been worth to share this knowledge as a prerequisite to improve the understanding on cloud formation processes in these regions and in general, which finally would also be a benefit for climate modelling.

However, due to the extensive quality assurance of the lidar data that is needed after each transport of the lidar system, the submission was delayed with respect to the dramatic wildfires in Australia in the end of 2019/ beginning of 2020. We therefore regret, that after the intense discussion of these record-breaking smoke emissions in Australia in beginning of 2020, our scientific message was obviously not of high interest for the Referees anymore. Considering both reviewers' comments, we refrain from submitting a comment by comment response and a revised version.
* * *